# Bio-Physics Approach to Urinary Incontinence Disabilities

**DOI:** 10.3390/ijerph191912612

**Published:** 2022-10-02

**Authors:** Loris Prosperi, Giovanni Barassi, Maurizio Panunzio, Raffaello Pellegrino, Celeste Marinucci, Antonella Di Iulio, Antonio Colombo, Marco Licameli, Antonio Moccia, Mario Melchionna

**Affiliations:** 1Center for Physiotherapy, Rehabilitation and Re-Education (Ce.Fi.R.R.), Venue “G.d’Annunzio” University of Chieti-Pescara, 66100 Chieti, Italy; 2Center for Physiotherapy, Rehabilitation and Re-Education (Ce.Fi.R.R.), Venue Gemelli Molise Spa, Professionalization Didactic Center, “Catholic” University of Rome/Campobasso, 86100 Campobasso, Italy; 3Department of Scientific Research, Campus Ludes, Off-Campus Semmelweis University, 6912 Lugano, Switzerland; 4Department of Thoracic Surgery, “Santo Spirito” Civil Hospital, 65124 Pescara, Italy

**Keywords:** urinary incontinence, neuromuscular manual therapy, focused mechano-acoustic vibration therapy, pelvic floor muscles

## Abstract

Background: The terminology of urinary incontinence (UI) and pelvic floor dysfunctions is complex. It affects quality of life and daily activities in personal, social, and professional fields. Managing UI without pharmacologic therapies is effective with a low risk of adverse effects and a large benefit for increasing continence rates. The aim of this preliminary retrospective observational study is to evaluate the effectiveness of the association between manual therapy and focused mechano-acoustic vibrations in women with nonspecific UI. Materials and methods: A group of 15 incontinent women (mean age 59.5 ± 11.4), referred to the Physiotherapy Center, Rehabilitation and Re-education (Ce.Fi.R.R.), located at the University “Gabriele d’Annunzio” of Chieti-Pescara from January 2019 December 2021, were enrolled after medical examination. The women were evaluated at T0 (admission protocol), T1 (after 8 weeks), and T2 (after 12 weeks). All patients received the rehabilitation protocol twice a week for a total of 8 weeks (T1) and were reevaluated after 12 weeks (T2). Outcome measures were: the Pelvic Floor Disability Index, the Pelvic Floor Impact Questionnaire-7, and the MyotonPRO. Results: The analysis of MyotonPRO data showed no significant improvements in all parameters. The PFDI-20 and PFIQ-7 questionnaire results showed a significant reduction in scores between T0 and T2.Results over time of the ANOVA values confirming the significant differences in the PFDI-20 and PFIQ-7 questionnaire results but not in the MyotonPRO variables. Conclusions: Despite limitations and no significant results, this study demonstrated that the integration of manual and focused mechano-acoustic vibrations therapy improved the symptoms of UI and reduced its psychosocial impact. Further experience could be required to establish the place of this integrated approach in achieving long-term improvements in UI.

## 1. Introduction

The terminology of urinary incontinence and pelvic floor dysfunctions is complex. Indeed, urinary incontinence is defined by the International Continence Society as “the complaint of involuntary leakage of urine” [1]. Furthermore, it has a negative impact on quality of life, interfering significantly with the activities of daily life, in the personal, social, and professional fields [2]. Although physiotherapy, medication, or surgery could be used to treat UI and associated disorders, there remains a crucial stigma and feeling of humiliation attached to these conditions as well as their psychological impact. The clinical effects of UI on daily life could differ greatly, depending on the etiology, severity of the condition, and coping strategies. The psychological distress of women suffering from UI could see them stuck in a vicious cycle. Incontinence could be burdened with anxiety, feelings of embarrassment, and the constant fear that people could discover the condition. Indeed, depression has been shown to be more common in incontinent women [2]. Managing UI without pharmacologic therapies is effective, with a low risk for adverse effects and a large benefit of increasing continence rates. Unfortunately, evidence is insufficient to evaluate and compare the comparative effectiveness of no pharmacologic and pharmacologic treatments for UI. Evidence shows that no pharmacologic treatments were better than no treatment in increasing continence and improving UI. On the one hand, pharmacologic therapy provided control of UI’s symptoms, and on the other hand, many patients discontinued medication due to the adverse effects and high costs of treatment [3]. A non-pharmacologic approach to treating UI could be neuromuscular manual therapy, characterize by soft tissue manipulations. Its aim is to restore the balance of the structures composing the human body [4]. Furthermore, myofascial trigger points could be considered palpable hyperirritable nodules in skeletal muscle or fascia and they could appear in incontinent women, connected with pain and increasing discomfort. Electrical neuro-modulation, by reducing myofascial trigger points, could be a reliable alternative for improving the well-being of incontinent women [5,6]. Another therapy for effectively treating muscle system dysfunctions is vibrations, in particular, focused mechano-acoustic vibrations (VISS: Vibration Sound System). This kind of treatment shows a capacity to improve muscle metabolism, leading to the release of anabolic hormones and reducing the production of catabolic factors. Furthermore, it could stimulate higher centers of motor control by producing mechanical stimulations of specific mechanoreceptors and increasing the recruitment activity of muscle fibers [7]. Plus, the local application of focused mechanical-acoustic vibrations could normalize myofascial dysfunction [8]. Overall, in this study, we would like to promote an integrated rehabilitation treatment expressed through a “Biophysical Metric” approach that takes into account, first of all, the somatic dysfunction framed in the identification of the “Key Trigger Point”, stimulated with neuromuscular manual therapy techniques and through focused mechano-acoustic vibrations.

## 2. Materials and Methods

### 2.1. Design

The study was designed as a retrospective observational pilot study to evaluate the effectiveness of the association between manual therapy and focused mechano-acoustic vibrations in women with nonspecific UI.

### 2.2. Participants

A group of 15 incontinent women (mean age 59.5 ± 11.4), referred to the Center for Physiotherapy, Rehabilitation and Re-education (Ce.Fi.R.R.), located at the University “Gabriele d’Annunzio” of Chieti-Pescara from January 2019 December 2021, were enrolled after medical examination.

Inclusion criteria were: age range between 30 and 75 years old; diagnosis of UI (no specific type).

Exclusion criteria were: associated fecal incontinence, endocrine system pathology in the phase of metabolic decompensation, tumor diseases, severe psychiatric diseases, severe chronic diseases, pregnancy, and neurological diseases (especially those presenting genitourinary dysfunctions). All those who did not sign the informed consent to the study were also excluded. Women were evaluated at T0 (admission protocol), T1 (after 8 weeks), and T2 (after 12 weeks).

### 2.3. Rehabilitation Protocol

This study was conducted at the Physiotherapy Center, Rehabilitation and Re-education (Ce.Fi.R.R.), located at the University “Gabriele d’Annunzio” of Chieti-Pescara as part of the Centre of Sports Medicine. Written consent was obtained from all participants for the experimental procedure, according to the internal procedures defined by the ISO 9001-2015 standards for “Research and experimentation” (certificate of conformity—n. IT15/0304—of the management system for the provision of services of rehabilitative health care and for observational clinical studies). At the beginning of each session, patients were evaluated for myofascial trigger points, through a dermatome map and manually by a physiotherapist, to identify the key trigger points expressing the maximum somatic dysfunction present at that moment. Muscles linked with the pelvic floor were selected bilaterally. Specifically, the muscles that were chosen were the rectus abdominis muscle, adductor muscle, dorsal muscle, and gluteus maximus muscle. Specific areas of somatic dysfunction (key trigger points) identified were treated with manual therapy combined with focused mechano-acoustic vibrations. The focused mechano-acoustic vibrations were administered by using the Vibration Sound System (VISS) (European patent: Ep1824439–CE 1936 Certificate of Conformity—N HD 60114019—Unibell, Calco—LC, Italy). VISS therapy uses fast-moving air cones to produce a square-wave mechanical vibration which is transferred to the skin, passing through the surface layers and stimulating the mechanical receptors. The signals released by these receptors trigger interactions and biochemical processes and are able to change the course of different pathologies. The study protocol involved the use of VISS for fifteen minutes. Patients were asked to lay on a table in the supine position, completely relaxed. At this point, the cups were positioned at the level of the selected muscles. The frequency used in this case was 300 Hz, necessary for the stimulation of specific mechanical receptors and therefore for the normalization of muscles. All patients received the rehabilitation protocol twice a week for a total of 8 weeks (T1) and were reevaluated after 12 weeks (T2).

### 2.4. Outcome Measures

#### 2.4.1. Pelvic Floor Disability Index (PFDI-20)

The Pelvic Floor Disability Index is a health-related quality-of-life questionnaire for women with pelvic floor dysfunctions. According to symptoms, PFD-20 contains 20 questions divided into three scales: symptoms of genital prolapse, from 1 to 6 (POPDI-6); colorectal–anal symptoms, from 7 to 14 (CRADI-8); and urinary symptoms, from 15 to 20 (UDI-6). All questions involve 4 levels of dysfunction: not at all, somewhat, moderately, or quite a bit. The minimal score is 0 points (minimum dysfunction) and the maximum is 100 points (maximum dysfunction). The total score is the sum of the three blocks with a maximum score of 300 [9,10].

#### 2.4.2. Pelvic Floor Impact Questionnaire (PFIQ-7)

Created by Barber, the Pelvic Floor Impact Questionnaire-7 (PFIQ-7) is a shortened, less-detailed version of the Pelvic Floor Impact Questionnaire (PFIQ). It is another health-related quality-of-life questionnaire for women with pelvic floor dysfunction that contains the Urinary Impact Questionnaire (UIQ-7), the Pelvic Organ Prolapse Impact Questionnaire (POPIQ-7), and the Colorectal–Anal Impact Questionnaire-7 (CRAIQ-7). All questions involve four levels of dysfunction: not at all, somewhat, moderately, or quite a bit. The minimal score is 0 points (low implication) and the maximum is 100 points (maximum effect). The total score is the sum of the three blocks with a maximum score of 300 [9,10].

#### 2.4.3. Myoton-PRO

The MyotonPRO was used to assess muscle function and metabolic efficiency. It is a portable, non-invasive, highly technological device that can evaluate the mechanical characteristics of muscles. Analysis with the MyotonPRO was performed on selected pelvic floor muscles. The measurement was made by applying the needle-shaped pressure sensor of the instrument to the center of the specific muscle. Parameters considered were muscle basal tone and muscle elasticity/plasticity. The probability of error was set to 2% [11,12].

### 2.5. Statistical Evaluation

Data were collected at T0, T1, and T2 to determine how the patients, regardless of treatment, improved their conditions. Descriptive analyses were undertaken for all variables. A Student’s *t*-test paired sample was conducted to identify if there were significant differences between the T0 and T2 time points in all parameters measured. To test the effect of the intervention over time, a two-way analysis of variance (ANOVA) was conducted on outcome variables between-subject factor and time (T0, T1, and T2), followed by Tukey’s post hoc test. Further, the PFDI-20 and PFIQ-7 scales were tested by two-way repeated measures ANOVA. Differences were evaluated at a *p*-value < 0.05.

## 3. Results

The analysis of MyotonPRO data displayed no significant results for any parameters (Table 1). We tested the core of pelvic floor muscles, abdominal muscles, and back muscles, but not the diaphragm muscle. These muscles together attach to the pelvis and spine, creating stability throughout the body’s center. Of course, keeping the core and stability of these muscles is crucial to reduce the symptoms of urinary incontinence. They could influence each other’s symptoms. The MyotonPRO was used to assess muscle function and metabolic efficiency. The results showed how the rehabilitation protocol could change in terms of structure and efficiency so that the patients tested could represent this in their answers to the questionnaire. Changes in muscle structure allowed for better performance and stability of the pelvic floor system and an improvement in the women’s quality of life, according to the questionnaire asked about this topic. Better quality of structures means better performance and improvement of functional activities, as well.

The comparison between time T0 and T2 in the PFDI-20 and PFIQ-7 questionnaires showed a statistically significant reduction in values with an improvement in feeling in the observed group (Table 2).

Table 1 and Table 2 show the results over time of the ANOVA values that confirm the significant differences in PFDI-20 and PFIQ-7 questionnaire results but not in of MyotonPRO variables. No adverse events were recorded.

## 4. Discussion

The variety of pelvic floor dysfunctions leads inevitably to a wide choice of therapeutic possibilities developed over the years. Indeed, a lot of evidence has demonstrated how the training of the pelvic floor muscles is fundamental to obtaining an improvement in urinary dysfunctional symptoms. A recent study showed that achieving motivation could be necessary for the integration of physical exercise and cognitive learning [13]. Results showed that a bio-physics approach could be an effective and safe treatment for incontinence in women. Benefits could come from the specific integration of neuro-manual stimulation and VISS, which could help to achieve the right strength of pelvic floor muscles and coordination to be used in standard incontinence rehabilitation.

Thus, to help women who suffer from urinary incontinence, instead of using single therapy, it could be better to combine approaches that work with different pathways [14]. Even though the results were not statistically significant, they showed reductions in the myometric parameters. A reduction in myometric parameters is related to better muscle metabolism and performance. Indeed, low levels of muscle tone are usually associated with tissue relaxation [15]. Therefore, the muscle control of pelvic floor muscles and their relaxation are crucial in different kinds of pelvic floor disorders and urinary incontinence [16]. A reduction in muscle control and endurance could produce incoordination between muscle group agonists and antagonists. A low grade of balance could be associated with the instability of tissues and tendon control due to excessive flexibility [17]. Between the function and dysfunction of pelvic floor muscle control, there is a range of adaptations of the muscle system, so it is fundamental to keep this balance through neuromuscular stimulation. This approach prevented the stiffness and not-adaptation of pelvic floor structures [18]. Besides, the muscle tissue was able to influence the feedback and feedforward of the nervous system, producing reactions from afferent somatic information [19]. Neuromuscular manual stimulation could influence inputs and could help to restore balance in the body’s systems. The metabolic improvements observed could be attributed to VISS as well [7,8]. The questionnaire scores could be linked to the relaxing and rebalancing effects produced by neuromuscular therapy. Furthermore, they also could be related to the fiber strengthening induced by mechano-acoustic vibrations [8]. Finally, the comparison in the PFDI-20 and PFIQ-7 between T0 and T2 showed a significant reduction in uncomfortable feelings caused by incontinence, as evidenced, and consequently, an improvement in individuals’ psychosocial pathways [20].

*Weaknesses:* Firstly, the size of the group and the lack of an extended follow-up to observe the effectiveness of rehabilitation protocol. Secondly, a potential selection bias due to the convenience sampling of patients being drawn from those close at hand. Evidence showed weight loss was associated with recovery or improvement in the group of all types of UI [21]. In this study, we could not consider that parameter due to technical problems in assessing and monitoring this measure. Further studies with a large group would be necessary to define a relevant protocol to treat UI.

*Strengths:* Patients’ compliance during the treatment was excellent. Probably, the timing of the protocol and the relationship with the team increased patients’ motivation to access the program of rehabilitation training.

## 5. Conclusions

Despite limitations and nonsignificant results, this study demonstrated that the integration of manual and VISS therapy improved the symptoms of UI and reduced its psycho-social impact. Further experience could be necessary to determine the protocol of an integrated approach to achieve and keep the quality of life and functionality of the pelvic floor system in women who suffer from UI.

## Figures and Tables

**Table 1 ijerph-19-12612-t001:** Outcome measures statistical analysis in MyotonPRO parameters.

Variable *	Count	Mean	SD	*t*-Test *p*-Value *	ANOVA *p*-Value *
**Adductor Muscle D _T0L**	15	1.7	0.2		
**Adductor Muscle D_T2L**	15	1.6	0.2	ns	ns
**Adductor Muscle D_T0R**	15	1.7	0.2		
**Adductor Muscle D_T2R**	15	1.7	0.06	ns	ns
**Adductor Muscle F_T0L**	15	11.6	1.3		
**Adductor Muscle F_T2L**	15	11.2	0.8	ns	ns
**Adductor Muscle F_T0R**	15	11.2	1.6		
**Adductor Muscle F_T2R**	15	10.6	0.2	ns	ns
**Adductor Muscle S_T0L**	15	206.2	27.5		
**Adductor Muscle S_T2L**	15	204.4	25.7	ns	ns
**Adductor Muscle S_T0R**	15	204.9	37.4		
**Adductor Muscle S_T2R**	15	194	24.4	ns	ns
**Dorsal Muscle D_T0L**	15	1.8	0.4		
**Dorsal Muscle D_T2L**	15	1.7	0.4	ns	ns
**Dorsal Muscle D_T0R**	15	1.7	0.3		
**Dorsal Muscle D_T2R**	15	1.9	0.4	ns	ns
**Dorsal Muscle F_T0L**	15	13.8	2.2		
**Dorsal Muscle F_T2L**	15	12	2.3	ns	ns
**Dorsal Muscle F_T0R**	15	12.9	1.8		
**Dorsal Muscle F_T2R**	15	12.7	1.2	ns	ns
**Dorsal Muscle S_T0L**	15	234.5	23.8		
**Dorsal Muscle S_T2L**	15	238.8	29.2	ns	ns
**Dorsal Muscle S_T0R**	15	234.5	25.8		
**Dorsal Muscle S_T2R**	15	241.2	32.3	ns	ns
**Gluteus Maximus Muscle D_T0L**	15	2	0,5		
**Gluteus Maximus Muscle D_T2L**	15	2.15	0,09	ns	ns
**Gluteus Maximus Muscle D_T0R**	15	2.04	0.5		
**Gluteus Maximus Muscle D_T2R**	15	2.12	0.4	ns	ns
**Gluteus Maximus Muscle F_T0L**	15	11.51	0.9		
**Gluteus Maximus Muscle F_T2L**	15	11.07	1.1	ns	ns
**Gluteus Maximus Muscle F_T0R**	15	11.46	1.02		
**Gluteus Maximus Muscle F_T2R**	15	11.63	1.4	ns	ns
**Gluteus Maximus Muscle S_T0L**	15	231.2	29		
**Gluteus Maximus Muscle S_T2L**	15	235.5	29.5	ns	ns
**Gluteus Maximus Muscle S_T0R**	15	232.1	32		
**Gluteus Maximus Muscle S_T2R**	15	230.4	34.8	ns	ns
**Rectus Abdominis Muscle D_T0L**	15	1.9	0.5		
**Rectus Abdominis Muscle D_T2L**	15	1.9	0.3	ns	ns
**Rectus Abdominis Muscle D_T0R**	15	1.8	0.4		
**Rectus Abdominis Muscle D_T2R**	15	2	0.4	ns	ns
**Rectus Abdominis Muscle F_T0L**	15	12.83	1.4		
**Rectus Abdominis Muscle F_T2L**	15	12.93	1.6	ns	ns
**Rectus Abdominis Muscle F_T0R**	15	11.26	0.8		
**Rectus Abdominis Muscle F_T2R**	15	11.16	0.8	ns	ns
**Rectus Abdominis Muscle S_T0L**	15	231.2	29		
**Rectus Abdominis Muscle S_T2L**	15	235.5	29.5	ns	ns
**Rectus Abdominis Muscle S_T0R**	15	202.2	37.3		
**Rectus Abdominis Muscle S_T2R**	15	204.5	36.5	ns	ns

* *p*-value refers to the differences between T0 and T2 (Student’s *t*-test). * *p*-value refers to the repeated measures ANOVA results. SD: Standard Deviation. MyotonPRO parameters: Adductor muscle D-logarithmic decrement L(left)/R(right); Adductor muscle F-oscillation frequency L(left)/R(right); Adductor muscle S-dynamic stiffness L(left)/R(right); Dorsal muscle D-logarithmic decrement L(left)/R(right); Dorsal muscle F-oscillation frequency L(left)/R(right); Dorsal muscle S-dynamic stiffness L(left)/R(right); Gluteus maximus muscle D-logarithmic decrement L(left)/R(right); Gluteus maximus muscle F-oscillation frequency L(left)/R(right); Gluteus maximus muscle S-dynamic stiffness L(left)/R(right); Rectus abdominis muscle D-logarithmic decrement L(left)/R(right); Rectus abdominis muscle F-oscillation frequency L(left)/R(right); Rectus abdominis muscle S-dynamic stiffness L(left)/R(right).

**Table 2 ijerph-19-12612-t002:** Outcome measures statistical analysis in PFDI-20 and PFIQ-7 parameters.

Variable	Count	Mean	S.D	95% LCL of Mean	95% UCL of Mean	*p*-Value *	ANOVA *p*-Value *
**POPDI-6 T0** (Part of PFDI-20)	15	20.27	19.21	9.63	30.91		
**POPDI-6 T2** (Part of PFDI-20)	15	8.05	9.24	2.93	13.17	0.00132	0.000028
**CRADI-8 T0** (Part of PFDI-20)	15	9.79	9.43	4.56	15.01		
**CRADI-8 T2** (Part of PFDI-20)	15	3.75	7.39	0.34	7.84	0.00499	0.000635
**UDI-6 T0** (Part of PFDI-20)	15	27.77	18.94	17.28	38.26		
**UDI-6 T2** (Part of PFDI-20)	15	18.05	12.36	11.20	24.9	0.00015	0.000000
**PFDI-20 T0** **Total Score**	15	57.84	27.45	42.64	73.05		
**PFDI-20 T2** **Total Score**	15	29.86	16.84	20.53	39.18	0.00001	0.000000
**BLADDER T0 ** (Part of PFIQ-7)	15	33.65	21.33	21.83	45.46		
**BLADDER T2** (Part of PFIQ-7)	15	13.96	12.53	7.02	20.9	0.00030	0.000016
**BOWEL T0** (Part of PFIQ-7)	15	11.11	16.85	1.77	20.44		
**BOWEL T2** (Part of PFIQ-7)	15	4.44	8.71	0.38	9.27	ns	0.038834
**VAGINA T0** (Part of PFIQ-7)	15	20.95	14.92	12.68	29.21		
**VAGINA T2** (Part of PFIQ-7)	15	8.88	11.22	2.67	15.1	0.00183	0.000071
**PFIQ-7 T0** **Total Score**	15	65.71	46.14	40.16	91.26		
**PFIQ-7 T2** **Total Score**	15	27.3	28.19	11.68	42.91	0.00072	0.000015

* *p*-value refers to the differences between T0 and T2 (Student’s *t*-test). * *p*-value refers to the repeated measures ANOVA results.SD: Standard Deviation. Questionnaire parameters: PFDI-20, Pelvic Floor Disability Index; UDI-6, Urinary Distress Inventory; CRADI-8, Colorectal-Anal Distress Inventory;POPDI-6, Pelvic Organ Prolapse Distress Inventory; PFIQ-7, Pelvic Floor Impact Questionnaire.

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
