# Peer review of "Bio-Physics Approach to Urinary Incontinence Disabilities"

_ijerph, 2022, doi:10.3390/ijerph191912612_

Round 1
Reviewer 1 Report
I have a few questions about the research content.
1. Based on references, please explain the evidence that the muscles tested are related to urinary incontinence.
2. I can't understand the data presented in the result table. What properties of muscles did you measure with myotone, and what results did you get?
3. Please explain how you derived the correlation between the myotone measurement and the questionnaires.
4. Please provide relevant references for VISS therapy and, if possible, photos or drawings of this treatment.
Author Response
- Based on references, please explain the evidence that the muscles tested are related to urinary incontinence.
We tested the core in pelvic floor muscles : abdominal muscles, back muscles, and diaphragm (the muscle that controls breathing). These muscles together attach to the pelvis and spine, creating stability throughout the body’s center. Of course to keep the core and stability of these muscles is crucial to reduce the symptoms in Urinary Incontinence.
- I can't understand the data presented in the result table. What properties of muscles did you measure with myotone, and what results did you get?
The MyotonPRO was used to assess muscles function and metabolic efficiency. The results showed how the rehabilitation protocol could change in term of structure and efficiency so the patients tested could represent in the answer of the questionnaire.
- Please explain how you derived the correlation between the myotone measurement and the questionnaires.
The MyotonPRO was used to assess muscles function and metabolic efficiency. The results showed how the rehabilitation protocol could change in term of structure and efficiency so the patients tested could represent in the answer of the questionnaire. Changing of muscle allowed better performance and stability and improvement of quality of life as the questionnaire asked in their topics. Changing of muscle allowed better performance and stability and improvement of quality of life as the questionnaire asked in their topics. Better quality of structures means better performance and improvement of functional activities.
- Please provide relevant references for VISS therapy and, if possible, photos or drawings of this treatment.
Pietrangelo, T.; Mancinelli,R.; Toniolo,L.; Cancellara,L.; Paoli,A.; Puglielli,C.; Iodice,P.; Doria,C.; Bosco,G.; D’Amelio,L.; Di Tano,G.et al. Effects of local vibrations on skeletal muscle trophism in elderly people: mechanical, cellular, and molecular events. Int J Mol Med 2009. 24 (4):503–512.
Barassi,G.; et al. Integrated Rehabilitation Approach with Manual and Mechanic-Acoustic V ibration Therapies for Urinary Incontinence. Adv Exp Med Biol. 2019; 1211:41-50. doi: 10.1007/5584_2019_436.

Reviewer 2 Report
The authors present a comprehensive approach to conservative treatment of urinary incontinence using manual therapy combined with the VISS device. The authors do not differentiate the type of urinary incontinence, dedicating their treatment to all types. They are aware of the significant limitations of the study resulting from the small study group and the short follow-up. The retrospective nature of the work and the lack of a control group also contribute to the low quality of the evidence. The authors achieved a significant improvement in the quality of life expressed with adequate questionnaires. However, it is not known whether similar results could be obtained with manual therapy alone or with VISS only. A recently published systematic review on conservative UI treatment demonstrated moderate to high certainty evidence of better cure or improvement with PFMT, electrical stimulation and cones compared to control. Moreover, weight loss was associated with recovery or improvement in the group of all types of urinary incontinence. (Cochrane Database of Systematic Reviews 2022, Issue 9. Art. No.: CD012337.DOI: 10.1002/14651858.CD012337.pub2). The authors of this study do not provide information on possible changes in body weight in the observed group of patients.
I propose to consider the study as a pilot study presenting a new device for training the pelvic floor muscles. Prospective randomized studies with a longer follow-up period are necessary to assess the quality of the described intervention.
Author Response
Thanks for your feedback. As we have mentioned in the text of the manuscript we have considered this study as a preliminary retrospective observational study to evaluate the effectiveness of the association between manual therapy and focused mechano-acoustic vibrations in women with no specific Urinary Incontinence. Indeed, weak points of the study are: firstly, the size of the group and a lack of an extended follow-up to observe the effectiveness of rehabilitation protocol. Secondly, a potential selection bias due to the convenience sampling of patients being drawn from those close to hand.
We have tried to understand how to change and improve managing of Urinary Incontinence with an integrated approach. We understand the importance of body weight but we couldn’t assess and motorize it in our study. Maybe we can discuss about this relationship in a new study.
Thanks for your feedback. As we have mentioned in the text of the manuscript we have considered this study as a preliminary retrospective observational study to evaluate the effectiveness of the association between manual therapy and focused mechano-acoustic vibrations in women with no specific Urinary Incontinence. Indeed, weak points of the study are: firstly, the size of the group and a lack of an extended follow-up to observe the effectiveness of rehabilitation protocol. Secondly, a potential selection bias due to the convenience sampling of patients being drawn from those close to hand.
We have tried to understand how to change and improve managing of Urinary Incontinence with an integrated approach. We understand the importance of body weight but we couldn’t assess and motorize it in our study. Maybe we can discuss about this relationship in a new study.
Thanks for your feedback. As we have mentioned in the text of the manuscript we have considered this study as a preliminary retrospective observational study to evaluate the effectiveness of the association between manual therapy and focused mechano-acoustic vibrations in women with no specific Urinary Incontinence. Indeed, weak points of the study are: firstly, the size of the group and a lack of an extended follow-up to observe the effectiveness of rehabilitation protocol. Secondly, a potential selection bias due to the convenience sampling of patients being drawn from those close to hand.
We have tried to understand how to change and improve managing of Urinary Incontinence with an integrated approach. We understand the importance of body weight but we couldn’t assess and motorize it in our study. Maybe we can discuss about this relationship in a new study.
